# Farrowing Pens for Individually Loose-Housed Sows: Results on the Development of the SowComfort Farrowing Pen

Inger Lise Andersen * and Marko Ocepek 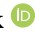

Department of Animal and Aquacultural Sciences, Faculty of Biosciences, Norwegian University of Life Sciences, 1432 Ås, Norway; marko.ocepek@nmbu.no
* Correspondence: inger-lise.andersen@nmbu.no; Tel.: +47-900-83-226

**Abstract:** The objective of the present paper was to discuss the design of farrowing pens and present the first production results of the "SowComfort farrowing pen" (SOWCOMF). The biggest difference between this pen and the traditional pen (TRAD) presented in the study, is that SOWCOMF contains a nest area equipped with a rubber mattress, floor heating, a rack for straw and no separate creep area. It was predicted that SOWCOMF would result in lower piglet mortality compared to TRAD due to a more stimulating and comfortable environment, and that the use of rubber mats in SOWCOMF would reduce the incidence of carpal joint lesions of the piglets. Results from both farms showed that percent mortality of live born piglets was lower in SOWCOMF than TRAD ($p = 0.004$), especially due to starvation ($p < 0.0001$) and other causes ($p < 0.0001$). In contrast, percentage of overlying was higher in SOWCOMF ($p < 0.0001$). The lower incidence of starved piglets in SOWCOMF than TRAD can possibly be explained by more sow-initiated communication with the piglets ($p < 0.001$). Most causes of mortality declined over consecutive batches. Percentage of piglets per litter without carpal lesions were significantly higher in the SOWCOMF than in TRAD ($p < 0.0001$), showing that rubber mats provide more protection of carpal joints.

**Keywords:** sows; SowComfort farrowing pen; piglet mortality; sow–piglet communication

## 1. Introduction

Piglet mortality caused by maternal crushing of piglets, many of which have no teat success, and starvation caused by sibling competition, increases with increasing litter size for most sow parities [1,2]. As the sows have become larger and longer during the selection process, the sows need more, sufficient space and especially wide enough pens for the sow to turn around and orientate within her nest area, both when performing nest-building behaviour [3,4] and when nursing [5]. Results from the PigSafe pen [6], suggest that the nest area should not be too large to minimize preweaning mortality. Comparatively, Cronin and co-workers [5] stated that the width of the nest area should be at least 2.2 m, to make it easier for the sow to orientate and nurse. Nestbuilding activity, sow-initiated communication with piglets outside the time of nursing and sow carefulness while being active in the pen are all traits that are highly important for piglet survival [7–9]. It is therefore important to create an environment that can stimulate those traits in a positive manner. To ensure that the sow can orientate and communicate with her piglets in an optimal way when moving around in the pen, sufficient space is needed. It is also important that there is a clear distinction between the nest/rest area and the activity/dunging area. There have been many attempts to develop farrowing pens that meet behavioural needs of sows. The Australian Werribee pen developed as early as in the late 1990s resulted in similar or even lower mortality rates than crates [10,11], and good results have also been achieved with the FAT2 pens in Switzerland [12].

The SowComfort farrowing pen (SOWCOMF) was developed with the aim of satisfying basic behavioural needs of the sow, thereby stimulating good maternal behaviour

and reducing the necessity of the farmer to interfere excessively around the time of farrowing. Key features to improve survival of piglets are access to nest-building material before farrowing [13–16], and the provision of sufficient space for the sow to turn around and orientate within her nest area, both when performing nest-building behaviour [4,10] and when nursing [5]. To encourage eliminating in the right place and that the sow is confident in her pen, it is important to have an open view in the slatted floor area that allows the sow the ability to see who enters the pen [10] and to enable some contact with neighbouring sows. Another important objective of a nest area design would be to increase the preference of sows to farrow in an area that contained specific features promoting piglet survival. For example, sloped, solid walls to lean against when descending from a standing posture to a resting position would be beneficial for the sow and reduce the likelihood of crushing [17]. Provision of floor heating in the nest area has previously been documented to help new-born piglets to dry faster, reduce heat loss shortly after birth and reduce latency to first suckling. This may result in as much as 7% higher survival rate compared to no floor heating [18]. Many farrowing pens are constructed with the assumption that new-born piglets are willing to leave the safe, soft, milk producing udder to go to a warm creep area. However, piglets under natural or semi-natural conditions would not leave the safety of the nest and their mother's udder during the first days after birth [19] as staying as close to the sow as possible would increase survival. Even when there is a high-quality creep area formed such as a hut with a small entrance, a thick layer of bedding and automatically controlled heaters, piglets still prefer to rest with their mother for the first two days [20]. Outside the time of nursing, the sow most commonly communicates to attract the piglets closer to her, but an increased time spent near the sow in this crucial period does not increase mortality [21]. Contrarily, the best "ticket" to survival for a piglet is to stay close to its mother for protection, warmth, comfort and ensure that an important meal is not missed. This is also why SOWCOMF does not have a separate creep area for piglets that the sow cannot access. Carpal joint lesions are quite often a challenge when pigs are suckling on a solid, concrete floor even when a large amount of sawdust is provided, and this may be enhanced in larger litters due to increased competition for teats.

The objective of this paper is to discuss the design and present the first commercial production results of an alternative farrowing pen, named the "SowComfort farrowing pen" (SOWCOMF). It was hypothesized that SOWCOMF would result in lower piglet mortality compared to TRAD partly because this pen design improves sow–piglet communication, and that piglet mortality would decrease with increasing number of batches in the new pen due to the farmer gaining more experience. Furthermore, it was hypothesized that the use of rubber mats in SOWCOMF would reduce the incidence of carpal joint lesions of the piglets shortly after birth.

## 2. Materials and Methods

### 2.1. Test Farms and Animals

Two commercial pig farms received financial support from Innovation Norway to proceed with building new farrowing pens in the farrowing section. Data on piglet mortality from the traditional pen system (TRAD) were collected before the installation of new pens in both farms. These farms were totally independent of each other but had the same traditional pens before installing the new pens. Each sow from all the different batches (batch is defined as a group of sows that are expected to farrow within approximately one week) was used only once. The interval between each batch was 8 weeks. Data were collected by one researcher in collaboration with the two respective farmers. In the 1st commercial farm, data from 119 healthy LY (Norwegian Landrace crossed with Yorkshire) sows of different parities and their litters were collected, of which 61 L were from three consecutive batches with the traditional pen system (TRAD; 20 L from batch 1, 20 L from batch 2, and 21 L from batch 3). After installing the SOWCOMF, data from the first 57 L from two consecutive batches (25 sows and litters in batch 1, 32 sows and litters in batch 2) were collected. On the second commercial pig farm, data during a longer period than in the first

commercial farm were collected. In the 2nd farm, data from 156 healthy LY sows and their litters were collected, distributed between three consecutive batches in TRAD (52 sows with litters per batch). From the SOWCOMF, production data from 343 healthy LY, distributed between 7 consecutive batches (49 sows and litters per batch) were collected. The sows were of different parities, of which 50% were primiparous and 50% were multiparous in both pen types. The data collection from TRAD started at the same time on both farms, but the data collection of SOWCOMF started around 9 months later in the second farm.

### 2.2. The Pens

The commercial version of SOWCOMF comprises two compartments: a nest area (a) and an activity/dunging area (b) (Figure 1). The nest area has solid side walls, sloped walls on three of the sides (specific design features are given in Figure 1 and a hay rack on the fourth wall allowing free access to hay or straw. The nest area has two zones (60 × 120 cm) with floor heating, one towards the back wall and one towards one of the sides walls in the nest area, and the concrete floor was covered by a 30 mm thick, hollow rubber mattress (Calma; www.kraiburg.com accessed on 1 May 2022). The activity/dunging area contained the sow feeder and drinker as well as a plastic slatted floor. The SOWCOMF nest area has solid side walls to provide a closed cave-like environment for the sow and piglets, affording the sow a visual barrier for privacy from neighbouring sow(s) whilst in the nest, and hence some sense of isolation from herd mates. The solid walls and the sloping panels were made of fibreglass, which is a long-lasting, hard-wearing material that is easy to clean and was recirculated from windmills.

A solid board at the bottom of the hay rack was preferred to avoid waste on the floor. Two other important features of the nest area are sloping panels along two walls and two, independently controlled, heated-floor zones (Figure 1). Sows prefer to lie against sloping panels when descending from a standing to lying posture (Figure 2; [17]). Hence, with the ability to control temperature in different floor zones, it is possible to influence where the sow lies relative to her litter. Twenty-four hours before expected birth, both heat zones in the floor of the nest area were set at 34 °C to provide the piglets with heat irrespective of the birth location in the nest area. Sows also prefer temperatures of around 35 °C at the time of birth [22], and then a substantial reduction in temperature when milk production starts. Twenty-four hours after birth, one heat zone (towards the end wall of the pen) was switched off to ensure that the sows still showed a preference to rest on this location of the nest area, even if they had reduced their temperature preference as their milk production increased. The other floor heat zone (towards the right short wall of the nest area) was maintained at 34 °C for most of the lactation period and reduced to 30 °C in the last week of lactation, to stimulate the piglets to choose this location for resting when not nursing. This wall is too short for most sows to lean against, and thus most sows preferred the area towards the back wall or the centre of the nest for resting and nursing. Most of the time, the sows and piglets could move around in the pen freely, but when the farmer wanted to inspect or handle the piglets it was possible to keep the sow and piglets separate by closing the entrance between the nest area and the dunging area (Figure 3).

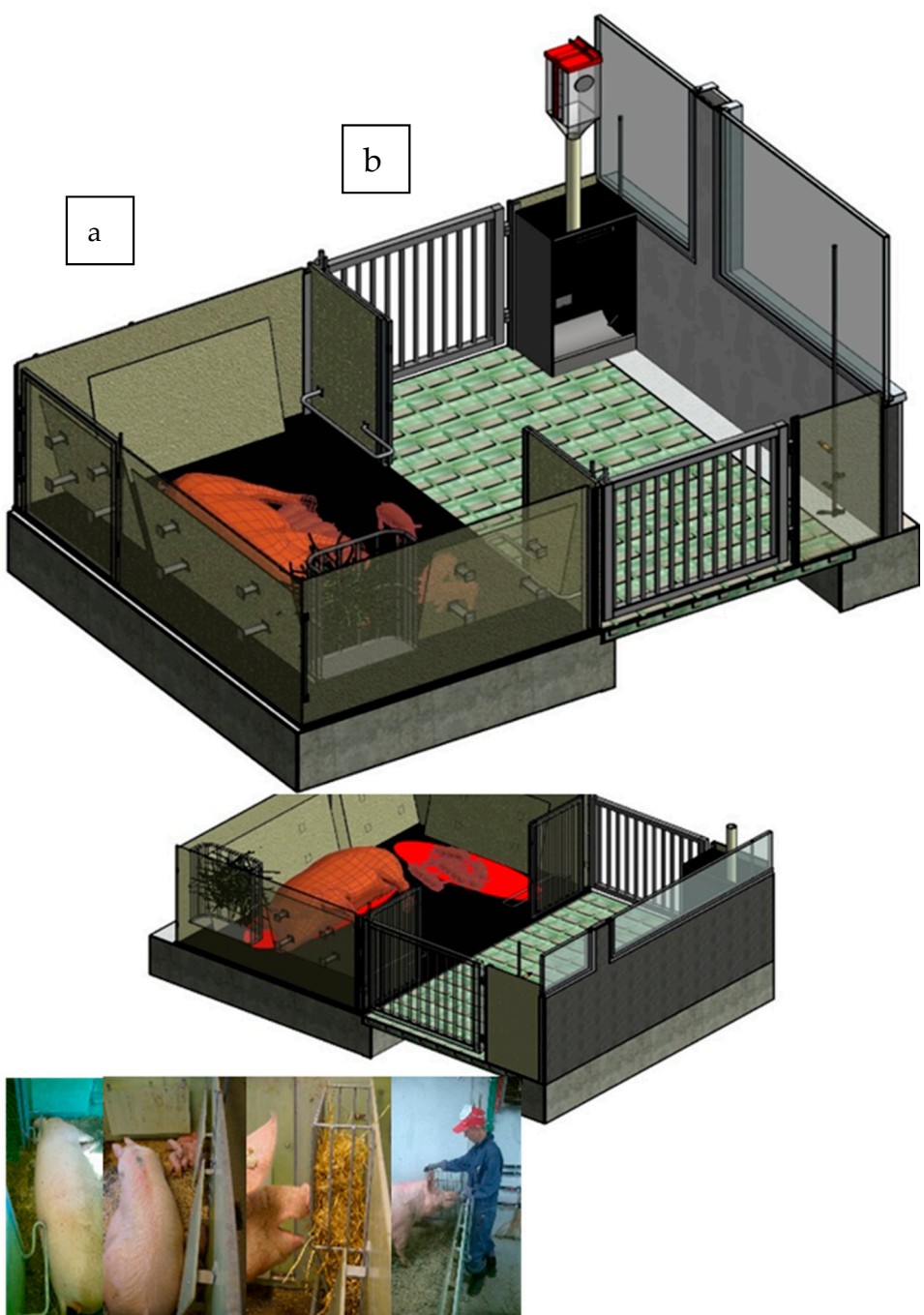

**Figure 1.** Design features of SOWCOMF showing a nest area (**a**) with sloped walls to minimize crushing and a rubber mat covering a floor that has two floor heating zones, a hay rack, an entrance to an activity area (**b**) with a plastic slatted floor that can be left open or closed. At the entrance, farrowing rails were used. The wall of the pen was only 120 cm high allowing for good contact between the farmer and the sows. The drawing was constructed by Elsbeth Morland in Fjøssystemer A/S (2017).

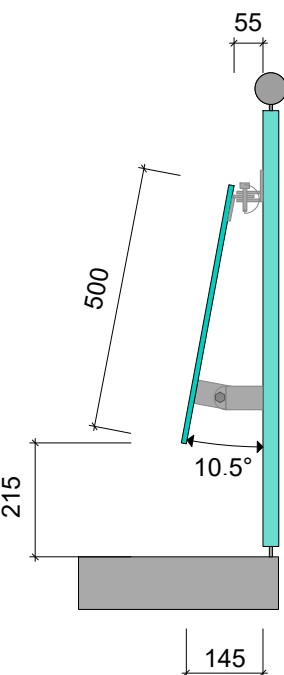

**Figure 2.** Details of the sloped wall in the pen. The angle between the solid wall and the sloped wall is given, and the other measures are given in centimetres.

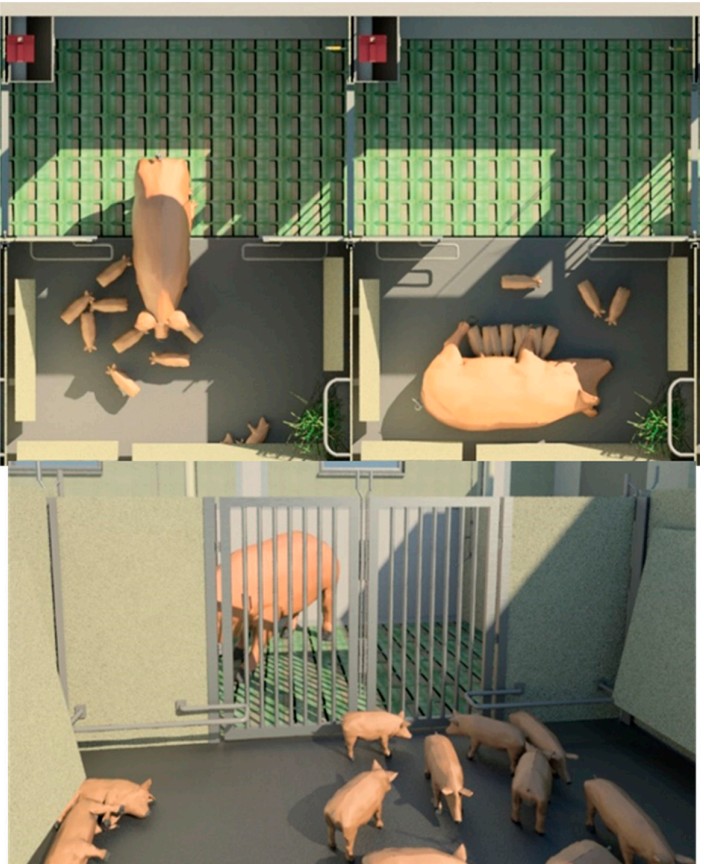

**Figure 3.** Schematic presentation of how SOWCOMF works in practice when the sow and piglets move around in the pen and when the door between the two areas are closed so that the farmer can handle the piglets without interference of the sow. The drawing was constructed by Elsbeth Morland in Fjøssystemer A/S (2017).

The TRAD pen used in both commercial pig farms, had a separate dunging and resting area, a traditional creep area for the piglets with a roof and infrared heaters. The dunging area of SOWCOMF was larger and the resting area smaller than in TRAD, and while SOWCOMF measured 7.7 m², TRAD measured 8.3 m² (Figure 4). TRAD did not have rubber mats, but rather the resting area had a solid, concrete floor with a generous amount of sawdust.

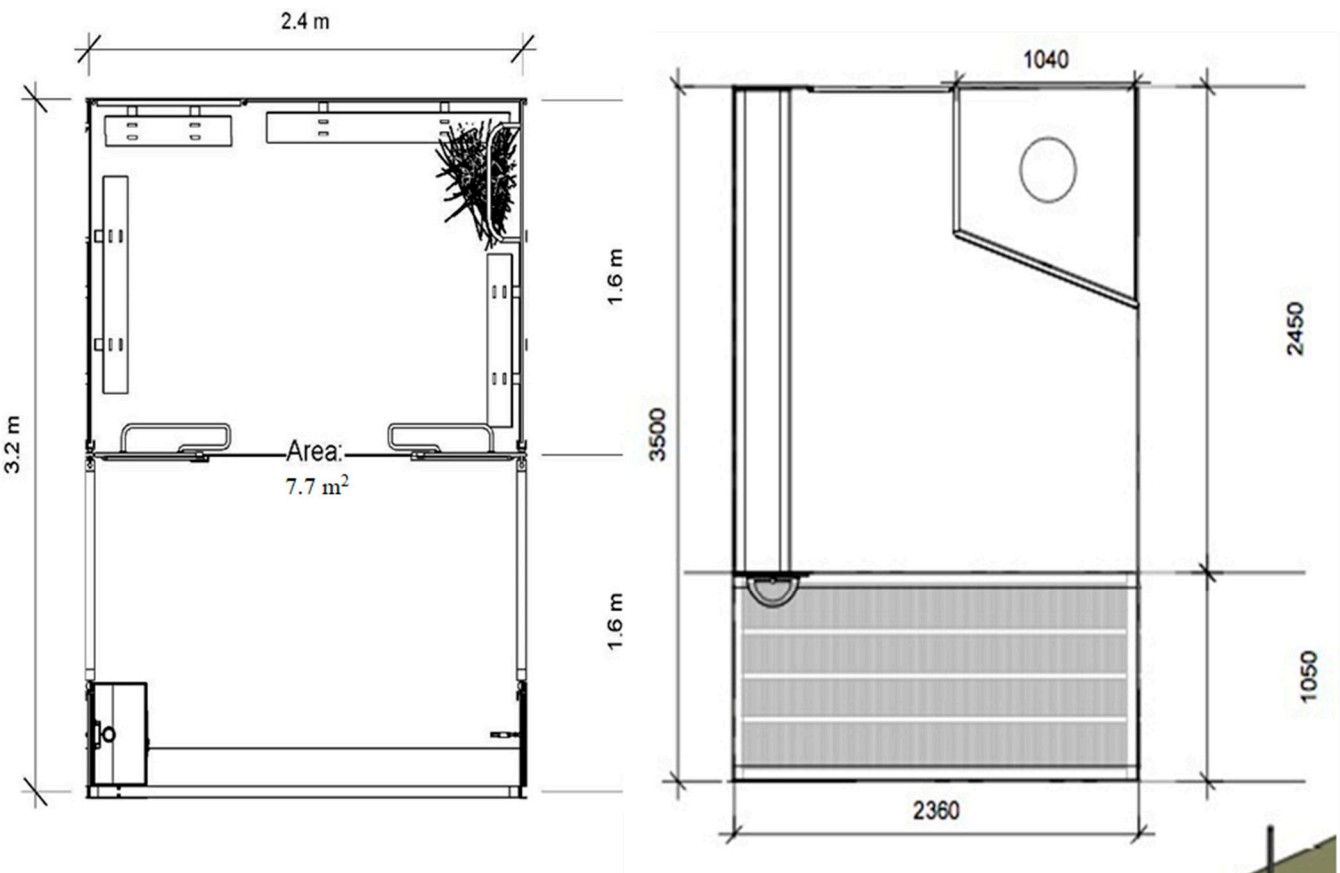

**Figure 4.** Overview of both SOWCOMF (**left**; measures given in metres) and TRAD (**right**; measures given in centimetres) with specific measurements.

### 2.3. Feeding and Management

In SOWCOMF, the sows were fed ad libitum with a standard concentrated, dry lactation diet from an automatic dispenser, placed in one corner of the activity area. They had free access to hay from a hay rack. The sows also had free access to water from a nipple drinker in the activity area and the piglets had free access to a nipple drinker placed below the sow drinker. Room temperature in both farms was kept at 18–20 °C during the entire period of data collection, and artificial light was kept on between 0730 and 1500 h. Regarding management, 80% of the births were attended from early morning to late in the evening, meaning that a person was present and could assist if any of the sows had birth problems, etc. The staff were not present during the night, but one person always conducted an inspection between 2200 and 2300 h. The piglets had an iron injection on day 2–3 post-partum. Pens were cleaned twice a day, and fresh sawdust was added in the lying area. The sows had free access to water from nipple drinkers. The feeding and management were similar in TRAD except that there was no hay rack in TRAD and thus no extra feeding of hay besides a limited amount of straw (less than 1 kg) used for nest building that was added at the time of farrowing. In TRAD, the sow was fed from a long trough with wet feed instead of an automatic feed dispenser with dry concentrate. The sows were given a small amount of hay from the hay rack during the entire lactation period, except for a short

period of 24 h before expected birth when the hay was replaced with free access to uncut straw for nest building.

At the time of birth, the layer of sawdust was 3–4 mm thick but later, it was reduced to just a thin layer not covering the mat. After birth, wet sawdust and straw were replaced by a thin layer of dry sawdust on top of the mat to ensure dry and hygienically optimal conditions for the neonates.

### 2.4. Postmortem Analysis and Assessment of Carpal Lesions

A post-mortem was conducted on-site on the same day as the pigs died or in the first morning after if they died during the night. This post-mortem could identify the most common causes of death, such as stillbirth, "no milk in the stomach" or "overlying". Variables used in the analysis were % mortality of liveborn, % stillborn (of total born), % overlain (of liveborn; either crushed by the sow when lying down or trampled on with fatal consequences), % of piglets that died without milk (of live born), and % that died of other causes (of liveborn). Stillborn piglets were identified by the colour of the lungs and whether they were inflated or not, and unclear cases were confirmed with a floating test to verify if there had been adequate oxygenation or no oxygenation of the lungs. The chest and stomach area of the dead piglets were opened to document whether the lungs were inflated and to assess whether there was milk in the stomach or not. Other causes could be that the piglets were born weak from the time of birth for instance due to prolonged farrowing and lack of oxygen, or due to malfunctions. Proportion of piglets per litter that had or had no lesions on their carpal joints were assessed by the experimenter and the farmer on day 2 post-partum in farm 1.

### 2.5. Sow–Piglet Communication

From the second farm, 10 sows (50% primi- and multiparous) from TRAD (last batch) and 12 sows from SOWCOMF (50% primi- and multiparous; last batch) were video recorded from the onset of farrowing until 12 h post-partum to count the number events with communication between mother and young. Communication is defined as grunting and sniffing with the snout directed towards one or more piglets within less than 20 cm. Only sow-initiated contacts were counted, covering situations where the sow actively oriented her snout towards one or more piglets combined with grunting or no grunting. Total number of sow-initiated interactions (involving sniffing with or without grunting) for the entire period was calculated. This information is crucial for piglet survival and in particular the incidence of piglets dying from hunger (no milk in the stomach) as sow–piglet communication is of great importance for nursing success (with milk let-down).

### 2.6. Statistics

The mortality variables and incidence of carpal joint lesions from each of the two farms were analysed separately due to slight differences in management routines and experience with the new pen system. In the analysis, a generalized linear model in SAS 9.4 (Genmod procedure) with Poisson distribution was used, including pen type and batch as fixed effects, and with number of liveborn piglets per litter as a continuous variable for the production data. Regarding production data, around 50% were primiparous and 50% were multiparous sows. All reported results are given as means and standard errors. Data on sow–piglet communication from the second farm were analysed using generalized linear model in SAS 9.4 (Genmod procedure) with Poisson distribution. Type of pen (SOWCOMF vs. TRAD) and parity were fixed effects in the model, and interaction between pen and parity was included in the model. Least-squares means test was used to assess differences between means.

### 2.7. Ethical Statement

The animals that were observed in these studies were in commercial farms following the Norwegian regulation for keeping pigs ([www.mattilsynet.no](www.mattilsynet.no) accessed on 1 May 2022),

which is stricter than most countries in the world. In Norway, crated sows were banned already in year 2000, and the pens presented in this paper are two of many alternative pens that are now legally on the market. The sows and piglets were not disturbed in any way, there was no interference with the farmers routines, and the SowComfort pen has the objective of improving the quality of the pen according to sow and piglet needs. The project with ID 01355422013 regarding testing of the pen was approved already on 14 December 2013 before the pilot version of the pen was constructed at the university farm, by Forsøksdyrutvalget (FOTS; www.mattilsynet.no/dyr;_dyrehold/dyrevelferd/forsoksdyr accessed on 1 May 2022).

## 3. Results

### 3.1. Effects of Pen Type, Batch, and Litter Size in Herd 1

Production results from the first herd showed that percent mortality of liveborn piglets was lower in the Sow Comfort Farrowing pen (SOWCOMF) than in the traditional pen (TRAD), and the causes of death between the two systems differed significantly (Table 1; Figure 5). There were more piglet losses due to starvation (no milk) and other causes in TRAD whereas overlying was more common in SOWCOMF (Table 1). Number of liveborn piglets did not differ significantly between pen systems in herd 1. All causes of mortality increased with increasing litter size irrespective of pen type, but so did the number of weaned pigs (Table 1). Mortality differed between batches in both pen types (Table 2). TRAD had the greatest number of liveborn piglets in the first batch and thus also the greatest piglet mortality, but still the number of weaned pigs were greater than in the two later batches (Table 1). For SOWCOMF mortality was greatest in the second out of the two batches most likely because this batch was assessed during the warmest summer month. In farm 1, percentage of piglets per litter without lesions on the carpal joints were significantly higher in SOWCOMF than in TRAD ($\gamma^2 = 96.5$; $p < 0.0001$; Figure 6), suggesting that the rubber mats provided more protection of the piglet carpal joints than concrete floor with sawdust. The percentage of piglets that exhibited carpal joint lesions declined significantly with increasing litter size ($\gamma^2 = 157.8$; $p < 0.000$), showing that larger litters are at higher risk for problems with carpal joint lesions.

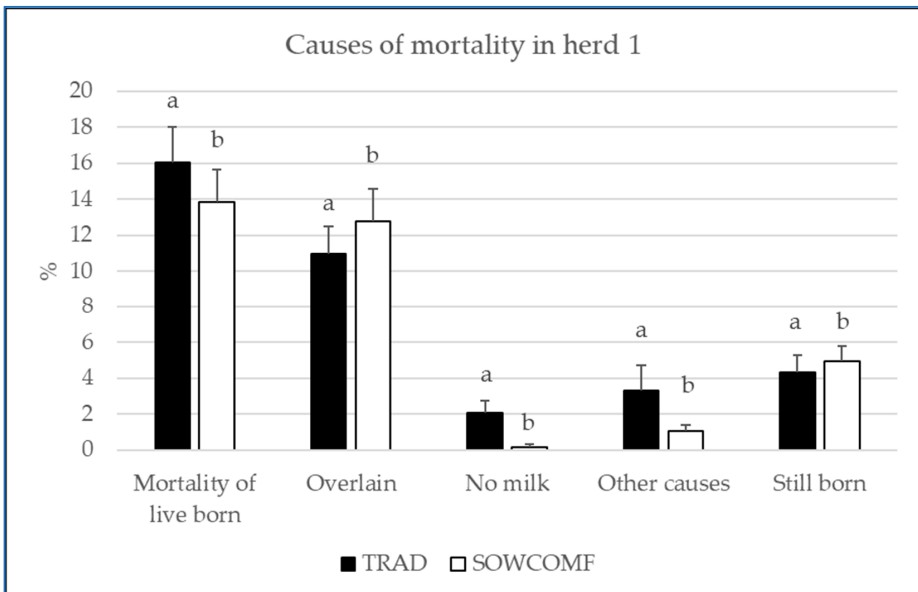

**Figure 5.** Causes of preweaning mortality (means + SE) in TRAD (Traditional: n = 61 L) vs. SOWCOMF (SowComfort farrowing pen: n = 57 L) in the first herd. Except for stillborn which was calculated as % of total born, the other variables were given as % of live born. For each variable, significant differences between the two types of pens (white and black bars) are denoted by different superscripts a and b.

**Table 1.** Effects of pen type, batch (3 batches with TRAD-traditional pen: n = 61 L and 2 batches with SOWCOMF- SowComfort farrowing pen: n = 57 L) and litter size on piglet mortality in farm 1. Except for stillborn which was given in % of total born piglets and number of weaned pigs, the other variables were given as % of live born piglets.

|  | Effect of Pen Type | | Effect of Batch | | Effect of Litter Size | |
|  | $\gamma^2$ | *p*-Value | $\gamma^2$ | *p*-Value | $\gamma^2$ | *p*-Value |
| --- | --- | --- | --- | --- | --- | --- |
| Mortality of live born, % | 8.5 | 0.0035 | 150.2 | <0.0001 | 105.2 | <0.0001 |
| Overlain, % | 108.3 | <0.0001 | 90.5 | <0.0001 | 55.7 | <0.0001 |
| No milk, % | 52.2 | <0.0001 | 375.4 | <0.0001 | 25.1 | <0.0001 |
| Other causes, % | 25.1 | <0.0001 | 41.5 | <0.0001 | 40.1 | <0.0001 |
| Still born, % | 0.9 | 0.3384 | 49.6 | <0.0001 | 151.1 | <0.0001 |
| No. of weaned pigs | 1.7 | 0.194 | 10.8 | 0.147 | 19.3 | <0.0001 |

**Table 2.** Production results (means ± SE) from the first farm during the three first batches of TRAD (Traditional: n = 61 L) followed by two consecutive batches in SOWCOMF (SowComfort farrowing pen: n = 57 L). Significant differences between each batch within pen type for all variables are denoted by different superscripts a, b, and c.

|  | TRAD | | | SOWCOMF | |
|  | Batch 1 | Batch 2 | Batch 3 | Batch 1 | Batch 2 |
| --- | --- | --- | --- | --- | --- |
| No. of live born | 14.4 ± 0.6 [a] | 12.4 ± 0.9 [b] | 12.2 ± 0.6 [b] | 12.7 ± 0.6 | 13.2 ± 0.7 |
| Mortality of live born, % | 19.6 ± 3.0 [a] | 12.5 ± 2.4 [b] | 9.3 ± 2.0 [c] | 11.8 ± 2.2 [a] | 15.9 ± 2.9 [b] |
| Stillborn, % | 4.3 ± 1.2 [a] | 4.1 ± 1.7 [a] | 8.9 ± 1.7 [b] | 4.9 ± 1.3 | 4.9 ± 1.2 |
| Dead without milk, % | 2.7 ± 0.9 [a] | 1.8 ± 1.1 [b] | 0.7 ± 0.7 [c] | 0.3 ± 0.3 [a] | 0.0 ± 0.0 [b] |
| Overlain, % | 12.6 ± 2.1 [a] | 8.8 ± 1.8 [b] | 7.5 ± 2.0 [c] | 10.5 ± 2.2 [a] | 15.1 ± 2.8 [b] |
| Other causes, % | 4.4 ± 2.5 [a] | 2.1 ± 1.3 [b] | 1.5 ± 0.9 [c] | 1.3 ± 0.5 [a] | 0.8 ± 0.4 [b] |
| No. of weaned piglets | 11.6 ± 0.7 [a] | 10.7 ± 0.8 [b] | 10.9 ± 0.5 [b] | 11.0 ± 0.5 | 11.0 ± 0.6 |

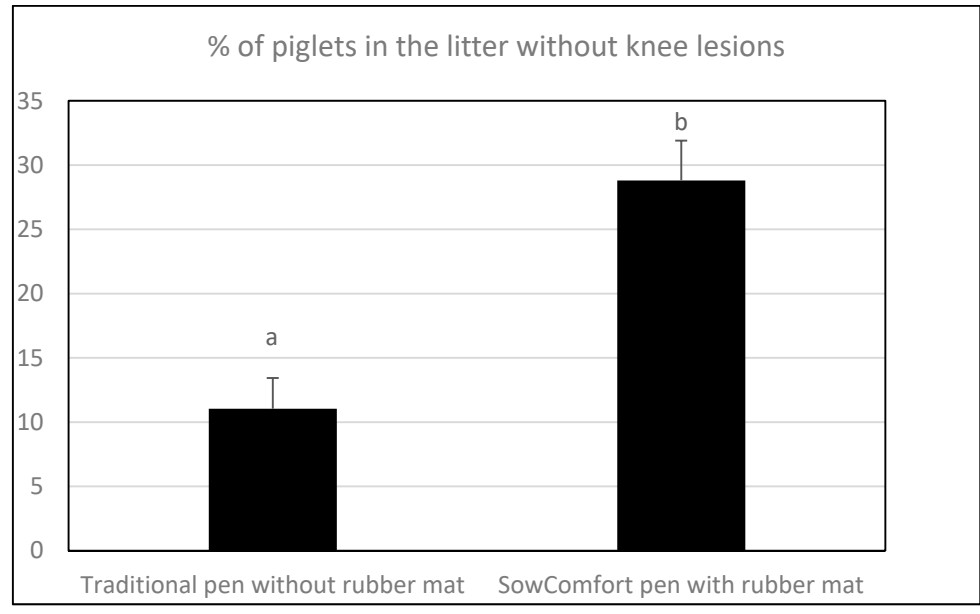

**Figure 6.** Percentage of piglets in each litter (means + SE) that did not have any carpal joint lesions on day 2 post-partum in farm 1 (n = 119 L). Significant differences between bars are denoted by different superscripts, a and b.

### 3.2. Effects of Pen Type, Batch, and Litter Size in Herd 2

In farm 2, where we could collect a much larger sample of sows for production data, mortality of liveborn piglets was significantly lower in SOWCOMF compared to TRAD

(Table 3; Figure 7). Number of liveborn piglets was high and similar for both pen types (TRAD: 15.1 ± 0.2; SOWCOMF: 14.4 ± 0.2). Causes of mortality in the two pens differed significantly (Table 3). While SOWCOMF resulted in lower mortality due to starvation (no milk in the stomach; Table 3; Figure 3), more piglets were overlain in this pen compared to TRAD. SOWCOMF also had significantly fewer piglets dying of other causes such as "born weak due to birth problems" or "other malfunctions", than in TRAD (Table 3; Figure 3). Percentage of stillborn (Table 3; Figure 7) and number of weaned pigs (TRAD: 12.6 ± 0.1; SOWCOMF: 12.4 ± 0.1) did not differ significantly between the two pen types. In a similar way as in herd 1, all causes of mortality increased with increasing litter size irrespective of pen type, but larger litters also resulted in more piglets weaned (Table 3). There were significant differences between batches for most variables (Table 3), except for number of weaned piglets that were stable across batches in both pen types. In the second farm, we collected data from three consecutive batches with TRAD. Batch 3 suffered the greatest piglet losses due to both starvation and overlying most likely because this was the warmest summer month of all the batches in the old farm building. The first seven consecutive batches with SOWCOMF showed that piglet mortality was significantly higher in the three first batches, and that production results improved already from batch 4 and onward (Table 4).

**Table 3.** Effects of pen type, batch (3 batches with TRAD-traditional: n = 156 L and 7 batches with SOWCOMF: n = 343 L) and litter size on piglet mortality in farm 2. Except for stillborn which was given in % of total born piglets and number of weaned pigs, the other variables were given as % of live born piglets.

| | Effect of Pen Type | | Effect of Batch | | Effect of Litter Size | |
|---|---|---|---|---|---|---|
| | $\gamma^2$ | *p*-Value | $\gamma^2$ | *p*-Value | $\gamma^2$ | *p*-Value |
| Mortality of live born, % | 8.5 | 0.004 | 150.2 | <0.0001 | 105.2 | <0.0001 |
| Overlain; % | 108.3 | <0.0001 | 90.5 | <0.0001 | 55.7 | <0.0001 |
| No milk, % | 52.2 | <0.0001 | 375.4 | <0.0001 | 25.1 | <0.0001 |
| Other causes, % | 25.1 | <0.0001 | 41.5 | <0.0001 | 40.1 | <0.0001 |
| Still born, % | 0.9 | 0.338 | 49.6 | <0.0001 | 151.1 | <0.0001 |
| No. of weaned pigs | 1.7 | 0.194 | 10.8 | 0.147 | 19.3 | <0.0001 |

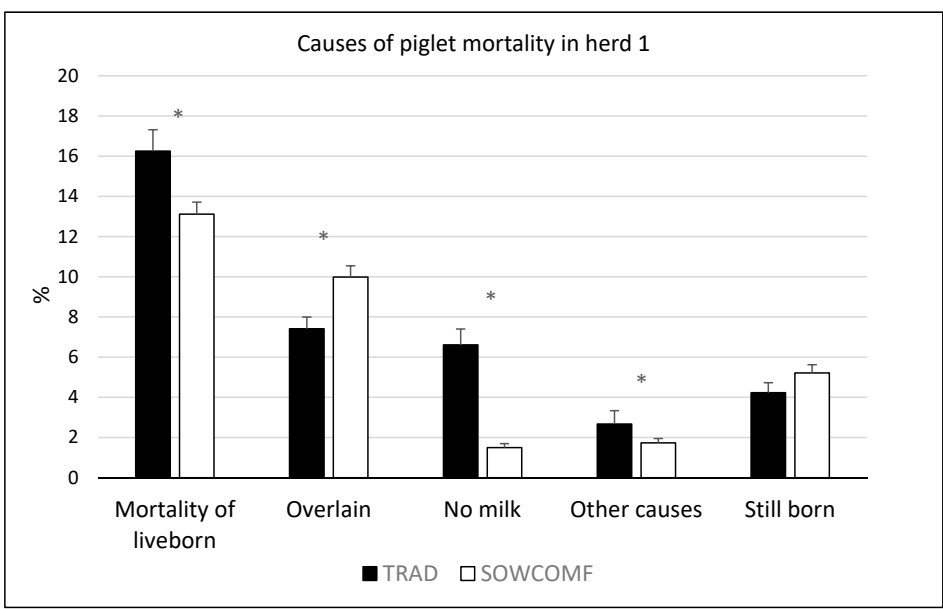

**Figure 7.** Causes of preweaning mortality (means + SE) in TRAD (Traditional; n = 156 L) vs. SOW-COMF (SowComfort farrowing pen; n = 343 L) in the 2nd herd. Except for stillborn which was calculated as % of total born, the other variables were given as % of live born. For each variable, significant differences between the two types of pens (white and black bars) are denoted by different *.

**Table 4.** Production results (means ± SE) from the 2nd herd during the three first batches of TRAD (Traditional: n = 156 L) followed by seven consecutive batches in SOWCOMF (SowComfort farrowing pen: n = 343 L). Except for number of liveborn and weaned piglets, the other variables are given in %. Significant differences between each batch within pen type for all variables are denoted by different superscripts a, b, c, and d.

| | TRAD | | | SOWCOMF | | | | | | |
|---|---|---|---|---|---|---|---|---|---|---|
| | Batch 1 | Batch 2 | Batch 3 | Batch 1 | Batch 2 | Batch 3 | Batch 4 | Batch 5 | Batch 6 | Batch 7 |
| No. of live born | 15.0 ± 0.4 | 15.4 ± 0.4 | 14.6 ± 0.4 | 15.0 ± 0.3 | 14.5 ± 0.3 | 14.3 ± 0.4 | 14.2 ± 0.2 | 15.6 ± 0.3 | 13.7 ± 0.3 | 13.8 ± 0.2 |
| Mortality of live born | 13.4 ± 1.6 [a] | 14.5 ± 1.4 [a] | 20.9 ± 2.3 [b] | 15.4 ± 1.6 [a] | 15.1 ± 1.3 [a] | 13.0 ± 1.8 [b] | 12.3 ± 1.6 [c] | 12.4 ± 1.3 [c] | 11.7 ± 1.6 [c] | 12.4 ± 1.8 [c] |
| Stillborn | 4.1 ± 0.9 | 3.9 ± 0.8 | 4.7 ± 0.9 | 2.9 ± 0.7 [a] | 4.8 ± 1.0 [b] | 5.5 ± 1.2 [b] | 5.5 ± 1.1 [b] | 4.6 ± 0.8 [b] | 6.5 ± 1.2 [b] | 6.3 ± 1.5 [b] |
| Dead without milk | 6.1 ± 1.2 [a] | 3.8 ± 1.0 [b] | 9.9 ± 1.7 [c] | 3.6 ± 0.7 [a] | 2.1 ± 0.6 [b] | 1.0 ± 0.3 [c] | 1.7 ± 0.6 [bc] | 0.2 ± 0.2 [d] | 1.3 ± 0.5 [bc] | 0.7 ± 0.4 [d] |
| Overlain | 5.6 ± 0.8 [a] | 7.8 ± 1.0 [b] | 8.8 ± 1.2 [c] | 9.8 ± 1.4 [a] | 12.1 ± 1.4 [a] | 10.7 ± 1.7 [a] | 9.1 ± 1.4 [a] | 10.7 ± 1.3 [a] | 7.8 ± 1.2 [b] | 9.9 ± 1.8 [a] |
| Other causes | 2.1 ± 0.7 [a] | 3.5 ± 1.2 [b] | 2.4 ± 1.4 [a] | 2.7 ± 0.6 [a] | 0.7 ± 0.3 [b] | 1.1 ± 0.3 [b] | 1.7 ± 0.6 [b] | 1.5 ± 0.5 [b] | 2.4. ± 0.8 [a] | 2.1 ± 0.7 [a] |
| No. of weaned | 13.2 ± 0.2 [a] | 12.9 ± 0.1 [a] | 11.6 ± 0.3 [b] | 12.2 ± 0.3 | 12.4 ± 0.2 | 12.5 ± 0.4 | 12.3 ± 0.2 | 13.8 ± 0.4 | 12.2 ± 0.2 | 11.7 ± 0.4 |

There were significantly more interactions between sows and their piglets through sniffing and grunting initiated by the sow directed towards one or more piglets in the period from onset of farrowing to 12 h post-partum (effect of pen type: $\gamma^2$ = 196.6; $p < 0.001$; Figure 8). There was no significant effect of parity per se ($\gamma^2$ = 2.2; $p = 0.130$), but there was a significant interaction between pen type and parity ($\gamma^2$ = 81.2; $p < 0.0001$), showing that there was no significant difference in number of contacts between sow and piglets between primi- and multiparous sows in TRAD, but that multiparous sows communicated more with their piglets than primiparous in SOWCOMF (Figure 8).

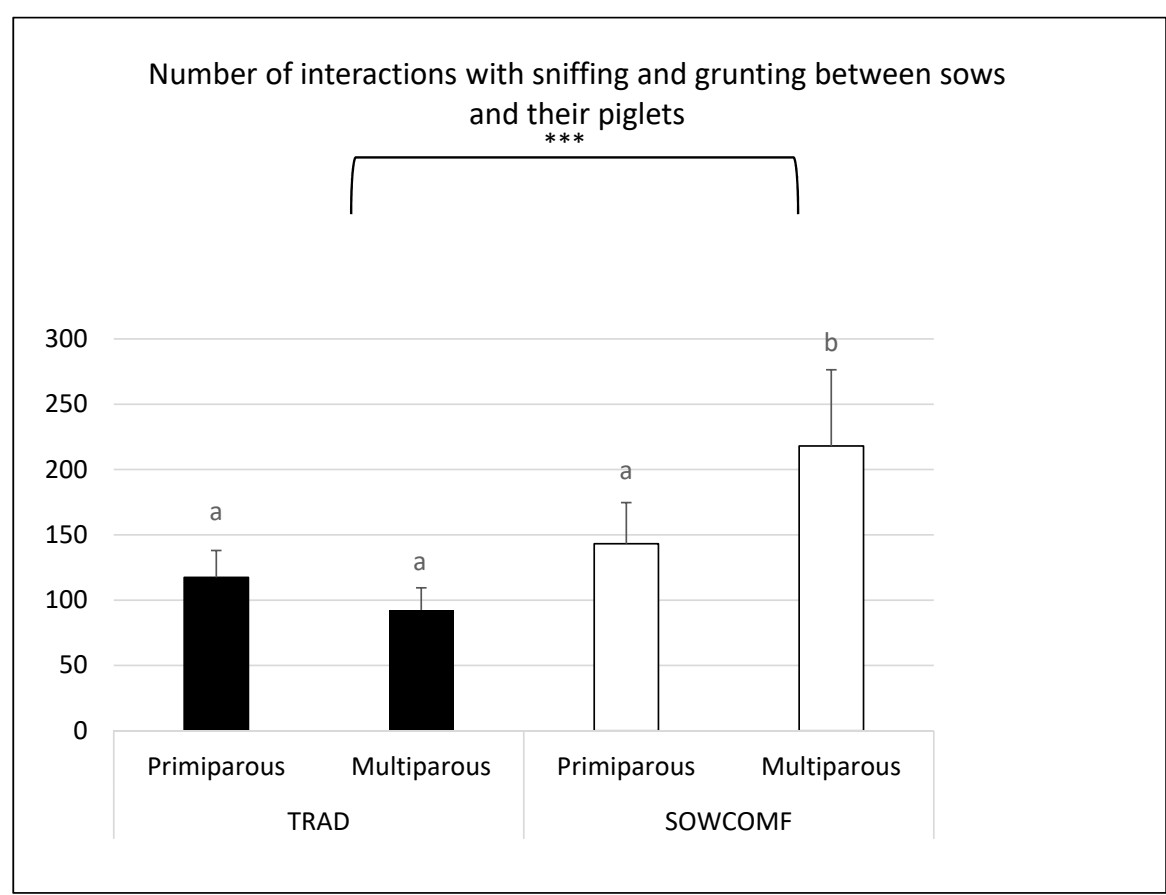

**Figure 8.** Mean + SE number of interactions with grunting and sniffing between sows and their piglets initiated by the sow approaching the piglets in the period from the onset of farrowing until 12 h post-partum. *** over the bracket: $p < 0.0001$: significant difference between the traditional pen type (TRAD: n = 10 sows) and the SowComfort (SOWCOMF: n = 12 sows) pen. Superscript letters ab: $p < 0.000$, denotes differences between primiparous and multiparous sows within the two pen types.

## 4. Discussion

As hypothesized, the sow comfort farrowing pen designed to meet behavioural needs of sows at farrowing, resulted in lower mortality of live born piglets than a traditional farrowing pen in both commercial farms, even though the traditional pen had greater floor space. Fewer piglets died of starvation and fewer died of other causes such as "born weak" or "with other malfunctions" in SOWCOMF, but more piglets died of overlying in this pen. The fact that fewer died of starvation or because they were born weak can be explained by the greater sow-initiated communication between mother and young in this pen, suggesting a greater maternal motivation and perhaps a more successful nursing. The latter still needs to be studied in more detail. Moreover, the fact that multiparous sows communicated more with their piglets than primiparous in SOWCOMF than TRAD, is a promising result as maternal motivation is expected to decrease with increasing parity, especially when litter size and maternal investment is great. To gain access to a teat soon after birth (i.e., short latency to suckle) increases the chances of piglet survival [10]. Floor heating in SOWCOMF and not in TRAD, may also reduce the likelihood of starvation as the piglets may dry sooner and hence have more energy to compete for teat access. This is in accordance with what is documented earlier by Malmkvist and co-workers [18]. In TRAD there was a creep area with an infrared heat lamp, but piglets do not start to use the creep area efficiently until day three post-partum in individually loose-housed sows [10]. Hence, the piglets are likely to have a higher heat loss shortly after birth, and some of them may become too weak to compete for a teat.

There was a higher percentage of piglets that died due to overlying by the sow in SOWCOMF than TRAD, showing that this remains a challenge when the litter size is large [1,2,11] even when sloped walls are provided. A pen without a separate creep area, providing the sow the freedom and absolute control over her piglets, requires good maternal skills. Overall, the greater the litter size, the lower the maternal motivation will become [22] and this creates an even more challenging task when the sow can decide how to interact with her piglets. To minimize the percentage of overlying, breeding organisations need to place less emphasis on litter size in the breeding goal compared to other traits such as maternal skills [23]. Still, it can be concluded that the design of SOWCOMF was more successful than TRAD as the overall mortality rate was lower. The fact that TRAD had a greater floor space, underpins the importance of focusing on the functional components to meet sow and piglet needs rather than floor space per se.

The results in the second farm showing that the percentage of overlain piglets was reduced substantially from batch one to seven, suggests that experience with a new pen system is important for the sows as well as the farmer. In Norway, mortality rates of liveborn piglets in individually loose-housed sows have become as low as 12 to 13% (InGris National data base, 2020), although the number of liveborn piglets are relatively high. It is not uncommon that a sow may bear 18 to 20 liveborn piglets even in early litters, which places a significant demand on the sow in terms of maternal investment. With large litters, it is the farmers management that is likely to have the strongest effect on piglet survival as the sow is not able to nurse and wean so many piglets on her own. This has been an ethical dilemma with pig breeding for many years and continues even though maternal traits and sow robustness have been improved. The traditional pen used in the present study is an example of a design that is commonly used for individually loose-housed sows in Norway. When discussing pen design, it is important to bear in mind, as documented in several papers [24–27], that mortality of live born piglets and the overall production results, are strongly affected by the farmers management, and more so than the pen design itself.

Another benefit with SOWCOMF was that the incidence of piglet lesions on the carpal joints was reduced, most likely owing to the rubber mat covering the floor of the nest area. By using rubber mats underneath the sawdust, the piglets avoid having direct contact with and rubbing knees against the concrete while suckling and fighting for teats, which is when the carpal lesions most commonly occur. In addition to resting comfort by providing a comfortable mat in the nest area, another key feature of the SOWCOMF

is the hay rack, allowing the farmer to provide the sows with straw for nest building or other types of roughage throughout the lactation period. This makes it easier to consider individual variation in the amount of material used. Providing sows with free access to relevant nest building material around 12 to 24 h before expected parturition is an excellent routine to improve the maternal motivation in sows, and long straw appears to stimulate nest-building the most and reduce the incidence of oral stereotypies [15]. Provision of straw also results in shorter farrowing duration, a lower percentage of stillbirths, and a lower frequency of negative communication towards piglets compared to when sows only have sawdust as litter in the pen [15,16]. Another important point to make is that the positive effects of nest-building are also dependent on enough space to move around and perform the behavior in a satisfactory way [25]. This ultimately means that provision of nest building material cannot compensate for the frustration of being confined. Larger pens with more bedding material, solid walls and not least with a deeper slatted floor area with open pen partitions results in a better dunging pattern and the cleanest pens [28].

In Norway, the current discussions about farrowing pens are rather focused on which design is the best for individually loose-housed sows, and the future trend is pointing to "from-farrowing-to 30 kg pens", where the piglets can remain and stay with the litter mates after weaning and the sow is moved. In this way, mixing and stressful environmental changes in the sensitive period are avoided immediately after weaning. Therefore, SOW-COMF has been modified into a larger pen more adjusted to the weaners, preferably with a minimum size of 8 to 9 m$^2$ (Figure 9).

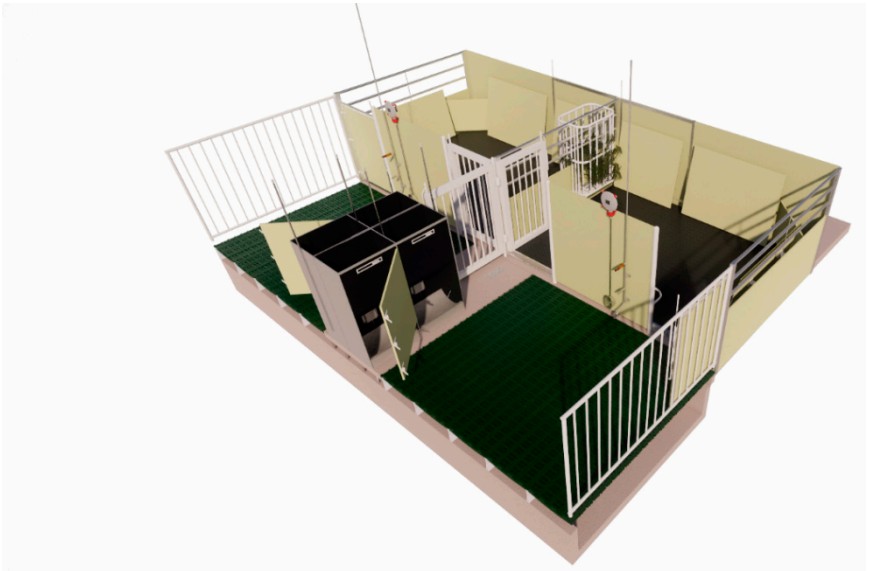

**Figure 9.** Slightly changed version of the SowComfort farrowing pen into a "from-farrowing- to-30 kg" pen where the litter can remain in their home pen until they are 30 kg. The pen has two feed dispensers and the entrance to the nest area is moved to one of the sides rather than in the middle to strengthen the walls and avoid an extra corner where piglets may choose to defecate and urinate. The drawing was constructed by May Helen Gryte in Fjøssystemer A/S (2019).

## 5. Conclusions

With an average of around 14 to 15 liveborn piglets, SOWCOMF with a nest area rather than a separate creep area, resulted in lower piglet mortality than TRAD with a separate creep area. Fewer piglets died of starvation or other causes than overlying, and the sow communicated more with her piglets in SOWCOMF than in TRAD. Finally, more piglets avoided carpal joint lesions in SOWCOMF than TRAD. This suggests that the qualities of this pen consisting of a comfortable nest area with floor heating, a hay rack and a mattress, and where sows and piglets can stay together, promotes nursing success and survival even in larger litters. After six batches, the mortality had declined from more than 15 to less

than 12%, suggesting that this pen system bears great potential for reducing mortality with increasing experience from sows and farmers and that the results with this pen depend on effective management.

**Author Contributions:** Ideas, pen development, analysis and original draft preparation: I.L.A. Writing of the paper and discussion of recent developments: M.O. All authors have read and agreed to the published version of the manuscript.

**Funding:** The experimental part of this paper was financially supported by Innovation Norway, Agriculture section (https://www.innovasjonnorge.no/ accessed on 1 May 2022). In-kind effort and pen construction was conducted by Fjøssystemer A/S.

**Institutional Review Board Statement:** Not applicable.

**Informed Consent Statement:** Not applicable.

**Data Availability Statement:** The data presented in this study are available on request.

**Acknowledgments:** The authors are grateful for the collaboration with Fjøssystemer A/S for their technical skills in developing the SowComfort Farrowing pen with us, and we especially give thanks to Elsbeth Morland, Harald Bore and May Helen Gryte for their direct involvement. We would also like to thank Innovation Norway and the two farmers for volunteering to test the commercial version of the SowComfort farrowing pen. Moreover, would like to acknowledge the staff at May Farm Piggery and Research Farm, and Greg Cronin, Faculty of Veterinary Science in Camden, University of Sydney, Australia for testing of the pilot version of the pen.

**Conflicts of Interest:** The authors declare no conflict of interest.

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
