# Peer review of "Farrowing Pens for Individually Loose-Housed Sows: Results on the Development of the SowComfort Farrowing Pen"

_agriculture, doi:10.3390/agriculture12060868_

Round 1

Reviewer 1 Report

1.       In the introduction, (Line 76   79), you present the objectives, but those are not clear, I respectfully suggest focusing on your hypothesis

2.       Line 78: You wrote, “designed to meet the basic behavioral needs of the sow at farrowing”, but this is not tested on the paper.

3.       Lines 80 – 83 moves to discussion section

4.       Figure 1 is missing

5.       Figure 2 needs more description, maybe include which wall… Please describe units of the dimensions

6.       Figure 4. Please indicate which is “sowconf” and which is “trad”. In the left graph the units are in m2, but in right side it does not have unit, seems ar?

7.       Line 206:  I don’t understand why you use just 10 o 12 sows, 50% primi and multiparous, but in line 225 mentioned that “there were no secure information about sow parity” please be consistent.

8.        Please include P values when report significant differences.

Author Response

1. line 76 and 79: Sentence about two farms with financial support is deleted from the aim. 

2.Line 78: "designed to meet behavioural needs...." is deleted

Overall, the objective now is focused on the hypotheses.  The reviewer is completely right in adressing this point.

3. l 80-83 is now deleted from the objectives

4. Figure 1 must was lost from the last document, and we are so sorry for this. The figure is now included in the new document.

5. More explanation is included in the figure text of figure 2

6. Corrections are made in the text of figure 4.

7. l 206, l225 Info is changed according to the reviewers request.

8. The system we have used is to refer to the table with statistical values when reporting significant differences for the results presented in figures and tables, but for those that are presented in the text only, we also have P- and F-values in brackets. Insted of inserting P-values in the text when reporting significant differences for the ones that are in tables, we have included more references to the stat-tables. I hope the reviewer find that satisfactory.

Finally, thank you for being so patient with our manuscript

Reviewer 2 Report

The paper deals with an interesting topic and totally up to date to provide farmers with solutions for free farrowing (which in Europe may become mandatory under The End of the Cage Age initiative). The paper is clearly written, with nice figures and with a discussion supported by the results obtained.

There are only minor comments with very specific issues:

Figure 1 does not appear in the pdf that this reviewer could download. Only the Figure Caption appears.

It is not totally clear to the reviewer how stillborn piglets were evaluated as compared to “piglets with no milk in stomach”.

Line 224-225, it states that that no information/security on the productivity data on whether it was primiparous or multiparous sow. Again this creates a bit of confusion. Sows were recorded, 50% primiparous and 50% multiparous to evaluate behaviour, and results are discussed accorging to that (lines 320-321). Why productivity data was not available?

Figure 5 and 7 display the overall causes of mortality for the three batches. However, differences in batches were found. Could this overall causes be biased by this batch effect?

Author Response

  1. Figure 1 was missed from the last MS. Sorry for this error. This is now included
  2. Stillborn piglets vs piglets with no milk: more details are included about this in the description of the M/M
  3. 224-225: confusion about parity is corrected, same comment as reviewer one
  4. Figure 5 and 7. There is always effects of batches in these types of studies. This is also why we prefer to have several batches, but  the main effect should not be biased by the batch effect in the present study, as the main finding are quite clear even in our small data set. However, it would be more elegant to have more batches. This was unfortunately not possible in the present iinnovation project. As mentioned earlier, we did not have resources t continue the data collection within the time period we had.

Reviewer 3 Report

Andersen and Marko have analyzed the design of farrowing pens and present the first production results of the “SowComfort farrowing pen”. This is an interesting topic. The experimental design is good. The writing and data presenting need to be improved. The following changes could improve the quality of the paper.

1.      Please add the important information to the footnote or figure legend of the tables and figures. Such as: abbreviations, replicates n=?, mean±SD/SE, ect.

2.      Please italic the P value throughout the paper.

3.      Lines 129-133, where is the figure 1?

4.      Line 263, please add “(3 batches with TRAD and 2 batches with SOWCOMF)” to the figure legends.

5.      Line 275-276, Space needed before and after “±”.

6.      The quality of the Figrue 6 need to be improved.

7.      Figure 7, using “*” to substitute the different letters to express the difference.

8.      Table 4, please using the writing way to present the table.

9.      Figure8, do not understand the meaning of “***” here?

10.  Figure 7, please improve the perspective of the figure.

Author Response

  1. mean + se is given in all tables and figures, Replicates are now included and abbreviations explained.
  2.  Italic for P-values in the text. Done!
  3. Figure 1 was no included by an error in the last MS. It is now inserted again.
  4. Batches and number of litters are added to the figure legends.
  5. Space before and after sign is added.
  6. A new and improved version of figure 6 is now included.
  7. We have inserted * to denote differences instead of letters.
  8. Table 4 is inserted again in the writing form.
  9. The meaning of ***: this is explained.
  10. Quality of figure 7 is improved.

This manuscript is a resubmission of an earlier submission. The following is a list of the peer review reports and author responses from that submission.

Round 1

Reviewer 1 Report

I suggest avoiding expressions in the first person

  1. Abstract

Please check to spell for   “loose-hosed”

  1. Introduction

I think that the paragraph about “outdoor systems”  is not relevant for this introduction, or need to be connected

In general, I suggest improving the coherence of this section, mainly the last paragraph. There is a mix of the methodology and presentation of the objectives. There are some phrases repeated. In my opinion, there are elements that could be useful in the discussion section and not here.

  1. Materials and Methods

It is difficult to understand the criteria to choose the sows on TRAD and on the new pens. You mention winter or summer, but it is not clear it was a criterion for the analysis. Also, I suggest explaining, what is a “batch” in this case.  Since in the results section, you present data relate to primiparous and second parity, I suggest mention, the number o sows evaluated in each parity

As well as in the previous section, I think you could improve the coherence of the descriptions,

Line 206: Please spell out the abbreviations a least the first time used in the text. i.e. LY ???

Line 240: Please check spelling “fiber-glas”

I suggest using better photos

Line 274:  Figure 1 does not correspond to the description

Line 251 – 260: Figure 2, please add a letter, then it is possible to relate it to the explanation on the text.

Line 312 – 316: I suggest describing feeders and other details for Sowconf in the previous paragraph

The statistical is not clear.  Which test did you use to compare means? How did you analyze batches?

  1. Results

Please, check if necessary to report F values.

Line 448: This analysis should be described on materials and methods, including the statistical procedure,

Since you analyzed the same variables on herd 1 and 2, if so, you should show those results

Are any data about carpal joint lesions for herd 2?

Table 1. Please add P values, (include n for each batch, if it is possible)

  1. Discussion

I suggest focusing on your hypothesis and related results.

Reviewer 2 Report

Dear authors,

thank you for taking the time and effort to revise and resubmit the manuscript "Farrowing Pens for individually loose-housed Sows: Results on the development of the SowComfort Farrowing Pen". You have improved the focus of the introduction and excluded the discussion on hyperprolific sows, as suggested. However, although there was some additional information added in the Material and Methods part I still found it lacking enough clarity and structure to fully assess the study design and statistcs. This lack of structure is also found in the results section where I miss a clear presentation of results in a comprehensive way. Tables including results for the comparison of TRAD vs. SOWCOMF would have been much more informative than the graohs and would have enabled a presentation of results for the different effects (e.g. parity) and for both herds. I already mentioned this in my previous rebiew. It is very difficult for the reader to follow design and results in these two sections. The discussion section focuses on some of the results in the beginning, but then focuses on more general aspects of farrowing systems. 

In contrast to the lack of clear presentation of study design and results, information outside the focus of the paper is overrepresented in my opinion, such as the rather extensive description of the development of the SOWCOMF-system from the prototype to the version used in this experiment or of the follow-up version for a "from-farrowing-to-30-kg"-system.

In addition to the above-mentioned issues I doubt that the manuscript has been thourougly revised with respect to language. There is still no consequent use of either American or British English nor of either past or present tense. Furthermore, some of the wording used does not seem appropriate and there are some sentences I find very hard to understand.

In summary, the revision did not take into account (or argue against) all of the points I have raised in my first review and still has serious flaws with respect to presentation of methods and results.